# The Endosomal Recycling Pathway—At the Crossroads of the Cell

**DOI:** 10.3390/ijms21176074

**Published:** 2020-08-23

**Authors:** Mary J. O’Sullivan, Andrew J. Lindsay

**Affiliations:** Membrane Trafficking and Disease Laboratory, School of Biochemistry & Cell Biology, Biosciences Institute, University College Cork, T12 YT20 Cork, Ireland; 115381596@umail.ucc.ie

**Keywords:** endosomal recycling pathway, vesicle trafficking, plasma membrane, Rab GTPases, cancer, neurological disorders, pathogen infection, small molecule inhibitors

## Abstract

The endosomal recycling pathway lies at the heart of the membrane trafficking machinery in the cell. It plays a central role in determining the composition of the plasma membrane and is thus critical for normal cellular homeostasis. However, defective endosomal recycling has been linked to a wide range of diseases, including cancer and some of the most common neurological disorders. It is also frequently subverted by many diverse human pathogens in order to successfully infect cells. Despite its importance, endosomal recycling remains relatively understudied in comparison to the endocytic and secretory transport pathways. A greater understanding of the molecular mechanisms that support transport through the endosomal recycling pathway will provide deeper insights into the pathophysiology of disease and will likely identify new approaches for their detection and treatment. This review will provide an overview of the normal physiological role of the endosomal recycling pathway, describe the consequences when it malfunctions, and discuss potential strategies for modulating its activity.

## 1. Overview of the Endosomal Pathway

### 1.1. The Endocytic Pathway

Endocytosis is the process by which cargo molecules are transported from the cell surface into the interior of the cell in membrane-bound vesicles. These cargoes are mainly transmembrane proteins and their ligands, which play a wide range of physiological roles including nutrient uptake, cell signalling, adhesion, and developmental regulation in response to morphogens. The human genome encodes approximately 5500 integral membrane proteins [1]. Many different endocytic pathways have been described, but they can all be broadly categorised into two mechanistically and morphologically distinct pathways: clathrin-mediated endocytosis (CME) and clathrin-independent endocytosis (CIE) [2]. The cytoplasmic domain of cargoes that undergo CME typically possess linear sequence motifs, or covalent modifications such as phosphorylation or ubiquitylation, that recruit adaptor proteins from the cytosol in a highly ordered manner, which leads to their packaging into clathrin-coated vesicles that are transported inside the cell. There are several diverse clathrin-independent endocytic pathways, and these are used by the cell to import membrane-bound receptors that mediate the transmission of signals from extracellular signalling factors to the nucleus. Furthermore, pathogens such as *Shigella* spp. and *Vibrio cholerae* subvert CIE pathways to gain entry into the cell. CME is the major endocytic pathway used by the cell to internalise cargo from the plasma membrane, with reports suggesting that up to 95% of endocytic vesicles are clathrin-coated [3]. Regardless of its route of entry, internalised cargo converges into a common early endosome (EE), a population of small vesicles and tubules, where they are sorted for onward transport to distinct cellular destinations. The EE is mildly acidic (pH 6.0–6.8), which facilitates the release of some ligands from their receptors. The majority of ligands that are internalised will undergo degradation by collecting in the lumen of the EE so that they can be sorted into late endosomes (LE) and finally into lysosomes where they are degraded. The receptors themselves can have a number of fates, such as transport to the *trans*-Golgi network, return to the plasma membrane, degradation in lysosomes, or transcytosis to the opposite membrane of polarised cells [4]. Transcytosis is critical for transport across epithelia, endothelia, and the blood–brain barrier. Ultimately, the central function of the endosomal network is to accumulate cargoes and arrange and then distribute them to their final destination.

### 1.2. The Recycling Pathway

The endosomal recycling pathway represents a dynamic system for sorting and re-exporting membrane components that have been internalized. Cargo internalised from the plasma membrane is sorted in the EE and can then either progress along the degradative pathway or be returned to the plasma membrane (Figure 1).

While the process of internalisation from the plasma membrane and the mechanisms that mediate transport along the degradative pathway are well understood, the machinery that regulates the sorting and recycling of cargo is less well characterised. Recycling back to the plasma membrane can occur directly from the EE (the fast recycling pathway) or indirectly via a distinct subpopulation of recycling endosomes (REs), often referred to as the endosomal recycling compartment (ERC), in a slow recycling pathway. The balance between endocytosis and recycling controls the composition of the plasma membrane [2], and it is emerging that disruption of this balance contributes to a wide range of diseases including cancer and neurodegenerative disorders [5,6]. Receptors internalized into EEs are frequently recycled back to the cell surface so that they can repeat the cycle of ligand-binding and internalization. Examples of cell surface proteins that undergo endosomal recycling include receptor tyrosine kinases (e.g., ErbB family members, IGF1R, FGFRs, c-Met), G protein-coupled receptors (e.g., Par1, chemokine receptors, beta-adrenergic receptors), cell adhesion molecules (integrins and cadherins), and carrier proteins such as the transferrin receptor (involved in iron uptake), low-density lipoprotein receptor (involved in cholesterol uptake) and Glut4 (transporter for glucose uptake) [7]. Such a dynamic system needs to be tightly regulated to ensure that the right cargo gets to the right location at the right time. Members of the Rab family of small GTPases play a central role in ensuring that this happens.

### 1.3. Traffic Control by Rab GTPases

Rab GTPases are ‘master regulators’ of intracellular trafficking. The human genome encodes over 60 different Rab GTPases, which control many diverse intracellular membrane trafficking pathways in the cell (Figure 2).

They have evolved a molecular switch mechanism to regulate all aspects of intracellular vesicle biogenesis and transport [8]. They cycle between a GTP-bound active conformation and a GDP-bound inactive conformation. This GTPase cycle is tightly controlled by a number of accessory proteins including guanine nucleotide exchange factors (GEFs), GTPase activating proteins (GAPs), GDP dissociation inhibitors, Rab escort protein (REP), and Rab geranylgeranyltransferase (Rab GGTase). Newly synthesised Rab is bound by REP, a chaperone that facilitates the addition of hydrophobic prenyl groups to the carboxy-terminus by Rab GGTase and the subsequent localisation of the Rab to its corresponding cell membrane. Once at its membrane, a GEF catalyses the exchange of GDP for GTP, which is present in the cytosol at 10 times the concentration of GDP. The now active Rab recruits a diverse repertoire of effector proteins which function to select cargo, form carrier vesicles from donor membranes, transport the vesicles along cytoskeletal tracks, and tether and fuse the carrier vesicles to the target membrane. Once it has performed its function, the Rab is inactivated by a GAP which promotes the hydrolysis of the GTP to GDP. GDI can remove the Rab from its membrane and shuttle it to adjacent membranes [9].

GEFs have very diverse structures, and it is likely that each Rab is activated by its own specific GEF, in some cases by more than one. More than forty GAPs have been identified, but they share more homology and have lower specificity for individual Rabs than GEFs. Effector proteins have diverse functions and include coat proteins, which mediate the formation of carrier vesicles; actin- and microtubule-based motor proteins that direct the transport of vesicles; phosphatidylinositol kinases that modulate the lipid composition of endosomes; and tethering factors and SNAREs that mediate attachment and fusion with the target membrane. Effectors are often multi-domain proteins, which allows them to coordinate different functions, and many can be recruited by several different Rabs [10]. In some cases, an activated Rab can recruit a GEF to a membrane, which in turn activates another Rab that functions downstream in the same trafficking pathway. Such Rab cascades orchestrate sequential steps along trafficking pathways, ensuring continuity in membrane transport. Rab11 subfamily members and also Rabs 14, 35, 8, 10, and 17 have all been implicated as regulators of the endosomal recycling pathway. The Rab11 subfamily is composed of three members: Rab11a, Rab11b, and Rab25. Rab11a and Rab11b share approximately 90% amino acid identity with each other and are widely expressed in different tissues. Rab25 is the most divergent of the three, sharing less than 70% identity with Rab11a and Rab11b, and its expression is restricted to epithelial cells. Rab11 is the most well-established regulator of endosomal recycling.

### 1.4. Rab11 Effector Proteins

The members of the Rab11 subfamily mediate their regulatory function by recruiting effector proteins, such as members of the Rab11 Family of Interacting Proteins (Rab11-FIPs), to membrane vesicles. These function in specific subdomains and have the ability to perform distinct roles within this recycling network. There are five members of the Rab11-FIP family—RCP, Rab11-FIP2, Rab11-FIP3, Rab11-FIP4, and Rip11—and all bind with high affinity to Rab11a, Rab11b, and Rab25 [11,12,13,14]. They all have a highly conserved ~20 amino acid Rab11-binding domain near their carboxy-terminus but can be divided into two classes based on the motif structure at their amino-termini. The class I FIPs have a C2 phospholipid-binding domain at their amino terminus and regulate endosomal recycling in interphase cells. The class II FIPs lack the C2 domain but possess a proline-rich domain and calcium-binding EF hands at their amino termini. Class II FIPs have a well-established role in delivering cargo to the cleavage furrow and intercellular bridge during cytokinesis. The class I FIPs also bind Rab14 [15]. It has been proposed that the FIPs occupy discrete domains of the endosomal recycling network and perform spatially and temporally distinct roles along this pathway [16].

The exocyst is a multi-subunit complex that plays an important role in the tethering of vesicles to the plasma membrane prior to their fusion and has been implicated in a wide range of cellular functions including exocytosis, cell growth, cell migration, cytokinesis, ciliogenesis, and tumorigenesis [17]. Rab11 interacts with the Sec15 subunit of the exocyst in a GTP-dependent manner [18], and this interaction is targeted by cholera and anthrax toxins in order to abrogate cadherin localisation at adherens junctions [19,20]. This leads to barrier disruption and vascular dysfunction during *Bacillus anthracis* infection and massive efflux of water across the intestinal epithelium in patients infected with *Vibrio cholera*.

Motor proteins physically transport cargo to their final destination along cytoskeletal filaments. This cargo includes protein complexes, mRNA, membrane vesicles, and organelles. There are three large superfamilies of molecular motors—kinesins, dyneins, and myosins. Kinesins and dyneins are long-range motors that transport cargo along microtubule ‘tracks’ in anterograde and retrograde directions, respectively. Myosins mediate short-range transport along actin filaments, typically just beneath the plasma membrane (PM). Rab11 recruits a number of motor proteins to REs that facilitate their transport to the PM. Active Rab11 recruits the kinesin motor Kif13A to the recycling compartment, where they cooperate to control the formation and transport of REs to the cell periphery [21]. Rab11 also associates indirectly, via Rip11, with Kif3B to mediate the entry of cargo into the slow recycling pathway [22]. Conventional kinesin, Kif5B, is required to transport Rab11-positive vesicles to growing pseudopods in macrophages [23]. Pseudopods are arm-like projections of the plasma membrane that surround particles during phagocytosis. Rab11 also binds the actin motor proteins, myosin Va and myosin Vb, which play a role in tethering and transporting REs along peripheral actin [24,25]. In hippocampal neurons, Rab11 and myosin Vb cooperate to transport AMPA receptor-containing REs into dendritic spines during long-term potentiation [26].

### 1.5. Overlap Between the Secretory and Recycling Pathways

The majority of membrane proteins, secreted proteins, and lipids originate at the endoplasmic reticulum (ER) and are conventionally thought to be transported from the ER to the PM via the ER–Golgi intermediate compartment (ERGIC), Golgi, and *trans*-Golgi network (TGN). However, a subset of newly synthesised proteins including E-cadherin, CFTR, and primary cilia proteins traffic through REs en route to the PM [27,28,29]. Recent work has identified WDR44, a Rab11 effector, as a key protein that connects the ER with REs in this unconventional secretory pathway. In HeLa cells, WDR44 localises to tubules that are closely aligned with the ER via direct interactions with the integral ER proteins VAPA and VAPB. These bridges may then facilitate the transfer of proteins into the recycling pathway. Inhibition of either WDR44 or Rab11 inhibits the transport of neo-synthesised E-cadherin, MMP14, and a mutant form of CFTR that causes cystic fibrosis, CFTRΔF508, to the PM [30].

## 2. Defects in Endosomal Recycling Have Been Linked to a Wide Range of Diseases

Dysregulation of the endosomal recycling pathway influences the onset of neurodegenerative diseases and the rare intestinal disease microvillus inclusion disease (MVID) and has been associated with an increased risk of cancer.

### 2.1. Parkinson’s Disease

Parkinson’s disease (PD) is a multi-system neurodegenerative disease and is the second most common age-related neurological disorder after Alzheimer’s. The hallmark symptoms include a resting tremor, bradykinesia, instability while standing, and rigidity. Before the presentation of motor symptoms, many patients will experience non-motor symptoms including depression, night disturbances, and loss of smell. These symptoms arise primarily due to the loss of dopaminergic neurons in the substantia nigra region of the brain [31]. Accompanying the loss of dopaminergic neurons is the formation of abnormal proteinaceous spherical aggregates called Lewy bodies in many of the remaining neurons. These form as a result of misfolding and abnormal aggregations of the protein alpha-synuclein (aSyn). Considerable evidence indicates that aSyn can be secreted and transferred from cell to cell, leading to the spread of the pathological events associated with PD. Rab11a interacts with aSyn and associates with the same intracellular inclusions. Overexpression of wild-type Rab11 reduces aSyn aggregation and increases its secretion, resulting in a reduction of the cytotoxic effects of aSyn [32]. Recently, four out of the nine hits from an unbiased shRNA-based screen of 1387 genes to identify modifiers of aSyn accumulation were found to be regulators of endosomal recycling (*Rab8b, Rab11a, Rab13,* and *Slp5*). Overexpression of Rab8b, Rab11a, Rab13, and Slp5 reduced the formation of aSyn inclusions and rescued aSyn-induced toxicity, and Rab11a and Rab13 enhanced a Syn secretion [33].

Ninety per cent of Parkinson’s cases are idiopathic, and 10% of cases are familial. Variants in a diverse range of genes have been linked to the inherited forms of PD and display a classical Mendelian pattern of inheritance [34]. Many of these are involved in endocytic membrane trafficking [6]. A new type of autosomal recessive early-onset parkinsonism (PARK20) has recently been described that is linked to mutations in the *Synj1* gene. Synaptojanin 1 plays a critical role in the control of the endocytic pathway, and its depletion leads to enlargement of EEs and inhibition of transferrin recycling, suggesting that defective membrane trafficking contributes to PARK20 pathogenesis [35]. Mutations in the *TMEM230* gene have been linked to an autosomal dominant form of familial PD. TMEM230 is a transmembrane protein that localises to REs in neuronal cell lines and to Lewy bodies in midbrain and neocortex sections from autopsy samples of patients with PD. The *TMEM230* mutations resulted in impaired vesicle trafficking in mouse primary neurons [36]. The findings described above indicate that defects in the endosomal recycling pathway are closely associated with the development of PD and are likely to play a key role in the pathogenesis of the disease.

### 2.2. Alzheimer’s Disease

Alzheimer’s disease (AD) is the most common neurodegenerative disorder, and its prevalence is rising due to the ageing world population. It is pathologically characterised by β-amyloid (Aβ) plaque deposition and neurofibrillary tangles of misfolded hyperphosphorylated tau protein [37]. These lead to the destruction of connections between brain cells and consequent memory loss, confusion, and difficulties in thinking. Aβ is secreted by neurons and arises from the proteolytic cleavage of amyloid precursor protein (APP) by two enzymes, β- and γ- secretase, in endosomes. Defects in the endocytic pathway are an early cytopathology in AD and precede Aβ deposition [38]. Rab11 interacts directly with presinilin-1, the catalytic subunit of γ-secretase [39], and β-secretase (BACE1) traffics between the PM and endosomes under the control of Rab11 [40,41]. Redirecting BACE1 away from REs leads to increased intracellular Aβ, whereas knockdown of Rab11a and Rab11b disrupts the endosomal recycling of BACE1, resulting in a consequent reduction of Aβ production [42]. Expression of a rare mutated form of presinilin-1, which is linked to familial AD, in cultured neurons causes Rab11 to accumulate in the soma and be excluded from the axon [43].

Pathological tau can spread throughout the brain and actively enter healthy neurons, where it acts as a template for the misfolding of normal tau, leading to the formation of neurofibrillary tangles. LRP1, a member of the low-density lipoprotein receptor (LDLR) family, was recently reported as mediating the entry of both normal and pathological tau into neurons [44]. Earlier work has shown that LRP1 undergoes endosomal recycling, which suggests that its cell surface levels are regulated by Rab11 [45].

Other links between the endosomal recycling pathway and AD include the statistically significant association of a Rab11 variant (rs117150201; T to G substitution in the 3′ UTR) with increased risk of late-onset AD [42] and the finding that mutations in the *SORL1* gene, which encodes the multifunctional intracellular sorting protein SORLA, have been associated with both early- and late-onset AD [46]. SORLA was recently found to colocalise with RE markers and mediate trafficking of cargo to the PM [47].

### 2.3. Microvillus Inclusion Disease

Microvillus inclusion disease (MVID) is a rare congenital disease that is characterised by life-threatening diarrhoea, partial sodium loss, and nutrient maladsorption during the first days and months of life. The enterocytes of patients have blunted or completely absent microvilli and an accumulation of intracellular secretory granules and inclusion bodies [48]. MVID has limited treatment options and a high mortality rate. It is caused mainly by inactivating mutations in the *MYO5B* gene, which encodes the actin-based motor protein myosin Vb. Duodenal biopsies taken from MVID patients with *MYO5B* mutations revealed the loss of Rab11a-positive REs from the apical cytoplasm of the enterocytes [49]. A single homozygous point mutation in *MYO5B*, which abolishes the interaction between myosin Vb and Rab11 and leads to MVID, has been discovered in the Navajo population in the United States [50]. Disrupted trafficking of REs that contain apical transporters, such as the sodium–hydrogen exchanger 3 (NHE3), to the plasma membrane has been proposed as the primary cause of the polarity defect and subsequent loss of microvilli in MVID, leading to maladsorption [51]. Whole exome sequencing of two patients that presented to the clinic with MVID symptoms but did not possess mutations in *MYO5B* identified premature stop codons in their *STX3* gene [52]. *STX3* encodes syntaxin 3, a member of the SNARE protein complex, and interacts with Rab11a-bound myosin Vb. They function together to regulate the trafficking of REs to the apical brush border of enterocytes [53].

### 2.4. Pathogen Infection

A pathogen is an infectious agent that causes disease in the host in which it resides, and many bacterial and viral pathogens have developed diverse mechanisms to subvert the host cellular machinery in order to successfully infect cells. The endosomal recycling pathway is one of the key cellular pathways that are frequently ‘hijacked’ by many important human pathogens, including members of unrelated viral (*Bunyaviridae, Paramyxoviridae, Herpesviridae, Filoviridae,* and *Orthomyxoviridae*) and bacterial (*Shigella* and *Salmonella*) families [54]. The viral lifecycle can be divided into four principal stages: entry, assembly, budding, and release. The recycling pathway has been subverted at each of these stages by different viruses, and Rab11 appears to be a key viral target. The endosomal recycling pathway can play a role in viral entry by regulating the presence of host cell receptors at the cell surface. For example, vaccinia virus binds to β_1_ integrin and CD98, with the trafficking to the PM of both being under the control of Rab11. Once internalised, vaccinia and some strains of Dengue virus are sorted into the recycling pathway and fuse with Rab11-positive REs prior to their uncoating in the cytoplasm [55,56]. Perturbation of Rab11 inhibits this process [55]. Newly synthesised viral ribonucleoprotein particles (vRNPs) from the *Bunyaviridae, Paramyxoviridae,* and *Orthomyxoviridae* families ‘hitchhike’ on Rab11-positive vesicles to sites of viral assembly or viral budding [57]. Evidence suggests that influenza A (IAV) vRNPs that exit the nucleus associate with the ER, where they are loaded onto Rab11-positive membranes and transported towards the cell surface, where they undergo final assembly and budding [58,59]. Rab11 vesicles also transport the HIV-1 accessory glycoproteins Vpu and Env to assembly sites. Nipah virus and Herpes-simplex virus-1 capsid proteins and Ebola virus VP40 protein use the endosomal recycling pathway to travel to the PM [57,60]. After their assembly into mature virions, viruses must exit the cell in order to spread and infect other susceptible cells, typically by budding directly into the extracellular milieu. Overexpression of mutant Rab11 or its effector proteins has been found to impair the release of *Paramyxoviridae* (respiratory syncytial virus [61,62]), *Orthomyxoviridae* (IAV [63]), and *Filoviridae* (Ebola [60]) viruses.

Coronaviruses such as SARS-CoV, MERS, and likely SARS-CoV-2, enter cells through the endosomal pathway and are uncoated in the acidic environment of the endolysosome, followed by release of the viral genome into the cytoplasm. Their genomic RNA is translated in the cytoplasm, which is quickly followed by the assembly of virions at the endoplasmic reticulum–Golgi intermediate compartment (ERGIC) [64]. Little is known about how the newly formed viral particles are delivered to the PM for budding and release; however, given the involvement of Rab11 and endosomal recycling in the life cycle of most other viruses, it is likely that it plays an important role.

Bacteria belonging to the *Salmonella* spp., *Shigella* spp., and *Chlamydia* spp. enter cells by macropinocytosis and end up in large membrane-bound vacuoles. In the absence of intracellular survival strategies, these vacuoles would fuse with lysosomes within 15 to 30 min, which would result in the digestion of their contents. However, these bacteria inject virulence proteins into the cytoplasm of the host cell that commandeer Rab11 in order to divert the vacuoles away from the endolysosomal pathway and thus protect them from destruction [65,66].

As can be seen, Rab11 and the endosomal recycling pathway are frequently subverted by intracellular pathogens to promote their own survival and reproduction. Manipulating endosomal recycling with small molecules may thus be a potential strategy for treating or preventing pathogenic infections. Indeed, small molecule inhibitors of the retrograde trafficking pathway (EE to TGN) have been found to protect mice from Shiga toxin, ricin, and several other intracellular pathogens [67]. These inhibitors are well tolerated by the mice, and since they target the host machinery rather than the pathogen directly, it is less likely that resistant pathogens will arise.

## 3. Dysregulation of Endosomal Recycling in Cancer

### 3.1. Rab GTPases Play a Role in Cancer Progression

Dysregulated trafficking of growth factor receptors and adhesion proteins (integrins and cadherins) is a common hallmark in malignant cells. This can be as a result of a number of abnormal mutations, amplifications, and deletions of genes that function to control the recycling pathway.

Rab25 is frequently amplified in breast and ovarian cancer [68], and its overexpression has been reported in prostate, lung, liver, glioblastoma, bladder, and gastric cancers [69]. In these cancers, Rab25 acts as an oncogene and promotes epithelial–mesenchymal transition (EMT) and metastasis. It interacts directly with β_1_ integrin and drives invasive migration by directing α_5_β_1_ integrin and EGFR to the leading edge of ovarian cancer cells [70]. However, the tumorigenic properties of Rab25 appear to be context-dependent. Loss of Rab25 is common in colorectal cancer and head and neck squamous cell carcinomas, where it appears to act as a tumour suppressor gene [71,72]. It is hypothesised that lack of Rab25 in these highly polarised cell types results in the loss of targeted delivery of cell surface proteins, possibly contributing to dedifferentiation of these cells. Loss of Rab25 is also common in basal subtypes of breast cancer. Indeed, ectopic expression of Rab25 induces apoptosis and suppresses the angiogenic properties of the triple-negative MDA-MB-231 breast cancer cell line [73]. The *Rab25* gene is often hypermethylated, and its expression switched off in these cancers [74,75].

As with Rab25, Rab11a has been reported to have both tumour-promoting and tumour-suppressing roles in cancer. Reduced expression of Rab11a is common in colon cancers and correlates with advanced stage and poorer survival. Low levels of Rab11a result in hyperactivation of the Hippo signalling pathway and an increase in the intestinal stem cell pool [76]. In contrast, overexpression of Rab11a positively correlates with stage and prognosis in non–small-cell lung cancer and leads to activation of YAP, a core component of the Hippo pathway [77]. Upregulation of Rab11a has also been observed in bladder and pancreatic cancer [78,79]. Rab11b has recently been reported to promote breast cancer brain metastases by controlling the surface expression of β_1_ integrin, which is required for successful interaction with the brain microenvironment [80]. A number of Rab11 effector proteins have also been implicated in cancer [81,82,83]. Rab14 and Rab35, which also play important roles in endosomal recycling, have been linked to cancer progression [84,85].

Another Rab that links endosomal recycling to cancer is Rab34. Primarily localised to endosomes and the Golgi apparatus, Rab34 plays a role in secretion, lysosome positioning, macropinosome formation, and ciliogenesis. A recent study found that Rab34 is highly expressed in aggressive breast cancers, binds directly to the tail of β_3_-integrin, and regulates the recycling of internalised αvβ_3_ integrin back to the PM. Knockdown of Rab34 inhibits cell motility, whereas EGF receptor activation induces the Src kinase-mediated phosphorylation of Rab34 on a tyrosine at its carboxy-terminus that leads to its subsequent translocation to the PM. The authors of the study hypothesize that αvβ_3_ integrin is internalised into endosomes upon activation. β_3_ integrin then activates Src already present on the same endosomes, which in turn phosphorylates Rab34. This phosphorylation event inhibits the lysosomal degradation of the integrin and enhances its recycling back to the PM to form new contacts with the ECM during migration and invasion [86].

### 3.2. p53 and Endosomal Recycling

p53 is a tumour suppressor protein often referred to as the ‘guardian of the genome’. It is lost or mutated in approximately half of all cancers, influencing cell cycle checkpoint controls and apoptosis. In addition to its anti-proliferative effects, p53 also contributes to other stages of cancer development such as cell migration and invasion [87]. Mutations in the *Tp53* gene most commonly occur in the DNA-binding domain between residues 102 to 292, 90% of which result in missense mutations that reduce the DNA binding capacity of p53. Some of the most common cancer-associated p53 mutations give rise to mutant proteins that are highly expressed that, in addition to losing transcriptional activity, have acquired a gain-of-function [88]. These gain-of-functions include the ability to drive tumour metastasis, partly due to their ability to disrupt the activity of p63, another p53 family member.

α_5_β_1_ integrin, the main receptor for fibronectin in the extracellular matrix, is a key contributor to cancer cell migration and invasion. Integrins are proteins that are activated through ‘inside-out’ signals, which is in essence an intracellular signal that promotes the binding of proteins like talin to the β-integrin tail. This allows the receptor to extend its conformation so that it will have high affinity for its ligands that are present in the extracellular matrix. Once a ligand has become bound to the receptor, it will cause ‘outside-in’ signals that recruit protein complexes to regulate cellular behaviour [89]. RCP physically links EGFR with β_1_ integrin and plays an important role in transporting α_5_β_1_ integrin- and EGFR- positive REs to the PM to promote invasion of tumour cells into fibronectin-rich 3D microenvironments [90]. p63 inhibits the association between RCP and α_5_β_1_ integrin; however, expression of mutant p53 disrupts p63 activity and relieves the inhibition. This leads to the acceleration of α_5_β_1_ integrin and EGFR recycling, elevated Akt activation, and a concomitant increase in the random migration of cancer cells [91].

p53 also controls the expression and secretion of many extracellular factors that are either soluble or contained in extracellular vesicles, such as exosomes. These have roles in cell–cell communication and ECM remodelling. More and more studies are reporting the involvement of mutant p53 in modulating the cancer ‘secretome’ to promote cancer invasion and metastasis [92]. RCP has been implicated in the exosome-mediated communication of mutant p53 gain-of-functions between tumour cells and fibroblasts in the stroma and to prime pre-metastatic niches in distant organs [93]. HSP90 is a chaperone protein that facilitates protein folding and plays an integral role in carcinogenesis, via the regulation of angiogenesis, cell proliferation, migration, invasion, and metastasis. The HSP90α isoform is secreted by cancer cells with a positive correlation between extracellular HSP90α (eHSP90α) levels and tumour malignancy. Recently published work shows that mutant p53 enhances eHSP90α levels in an RCP-dependent manner. The authors find that RCP interacts with HSP90α and acts as an adaptor to load Rab11-positive REs with HSP90α that will subsequently be secreted in exosomes [94]. The mechanism by which mutant p53 enhances RCP activity has not yet been elucidated, but identifying the relationship between the effects of wild-type p53 and mutant p53 on RCP expression will contribute to our understanding of how mutant p53 promotes endosomal recycling.

### 3.3. Receptor Tyrosine Kinases Undergo Endosomal Recycling

Receptor tyrosine kinases (RTKs) are key components of many signal transduction pathways that are involved in mediating cell-to-cell communication and control a wide range of complex biological functions including cell growth, motility, metabolism, and differentiation. They act as high-affinity cell surface receptors for a wide range of polypeptide growth factors, cytokines, and hormones, the binding of which induces highly regulated post-translational modifications, which control intercellular and intracellular communication. The human genome encodes fifty-eight RTKs, including epidermal growth factor receptor (EGFR), c-Met, vascular endothelial growth factor receptors (VEGFR), and fibroblast growth factor receptors (FGFR) [95]. Dysregulation of RTK signalling is frequently implicated in cancer development, and this aberrant activation is mediated by four key mechanisms: amplifications, gain-of-function mutations, chromosomal rearrangements, and autocrine activation [96]. Intracellular membrane trafficking plays an important role in determining the specificity and duration of signalling pathway activation [97]. Endocytic internalisation and the balance between degradation and recycling can profoundly affect the signalling properties of RTKs and is crucial for determining the net signalling output. The amount of RTKs at the cell surface that are available for ligand binding depends on the rates of these three trafficking processes. Upon internalisation, activated receptors are either sent to lysosomes for degradation or recycled back to the PM. The former results in the termination of the signal, whereas the latter supports sustained signalling. Furthermore, many activated RTKs stimulate different signalling pathways depending on whether they are at the PM or localised in endosomes.

#### 3.3.1. c-Met

When bound by its ligand, hepatocyte growth factor (HGF), c-MET activates wound healing and embryogenesis pathways in normal cells. However, aberrant activation of c-Met promotes carcinogenesis by stimulating a wide variety of signalling pathways, including the Ras/MAPK, PI3K/Akt, Src, Wnt, and JAK/STAT pathways [98]. c-Met is overexpressed or mutated in many tumour types and is an important therapeutic target. It is overexpressed in 20–30% of breast cancers and is correlated with disease progression, and high levels of HGF is associated with poor survival in ductal breast carcinomas [99]. In HER2-positive breast cancer, it is also often linked with poor survival outcomes [100] and is frequently associated with resistance to HER2-targeted therapies.

c-MET is subject to endosomal recycling, which is important for controlling the signalling pathways it activates. For example, it activates the JAK/STAT pathway from endosomes. A number of oncogenic c-Met mutations exhibit increased recycling and reduced degradative activity, resulting in the accumulation of activated receptor on endosomes and increased cell migration. Blocking endocytosis inhibits the tumorigenic properties of mutated c-Met [101]. c-Met interacts with RCP, and mutant p53 enhances its recycling via its stimulation of RCP. This results in the maintenance of activated c-Met and consequent pro-tumorigenic signalling. Thus, increased recycling of c-Met is an important driver of cancer progression.

#### 3.3.2. EGFR Endosomal Recycling is Frequently Dysregulated in Cancer Cells

The ErbB family of RTKs comprises four members: EGFR, HER2 (ErbB2), HER3 (ErbB3), and HER4 (ErbB4). These receptors are single-chain transmembrane glycoproteins that are structurally related. Ligand binding to their extracellular domain initiates homo- and hetero-dimerization between receptors, which is necessary to activate their intracellular tyrosine kinase domains and phosphorylation of their C-termini [102]. The EGFR is the prototypical member of the ErbB family and is essential for maintaining many cellular functions such as proliferation, survival, and migration. It has a number of ligands, including EGF, TGF-α, HB-EGF, and epiregulin. EGF receptors are rapidly internalised upon activation by both CME and CIE pathways, but different ligands can differentially regulate its signalling and trafficking. Once EGFR has been internalized, there is tight control of its endosomal distribution, i.e., whether it is degraded or recycled back to the plasma membrane. While the majority of signalling occurs at the plasma membrane, activated EGFR can still signal from endosomes, indicating that there are signalling pathways that depend on its intracellular trafficking. Furthermore, some reports suggest that EGFR can be transported on atypical trafficking pathways to the nucleus or mitochondria [103].

Dysregulation of the intracellular trafficking of EGFR family members plays a key role in oncogenesis by causing their mislocalization and upregulation, leading to enhanced signalling [104]. Activating mutations in the EGFR kinase domain are frequently found in non-small lung cancers (NSCLC), and these mutants preferentially transit into the recycling pathway where they activate the Src signalling pathway [105].

To gain a deeper insight into the mechanisms by which activation of the same RTK with different ligands can result in different signalling outcomes, Francavilla et al. utilised a systematic proteomics approach to study the cellular response to EGFR stimulation by EGF and TGF-α. They found that Rab7 phosphorylation and RCP recruitment are key regulators of EGFR signalling output. EGF, but not TGF-α, induced the phosphorylation of Rab7 on tyrosine 183 and promoted receptor degradation and thus signal termination. Conversely, TGF-α induced sustained activation of EGFR and promoted binding of RCP to the receptor and its subsequent entry into the endosomal recycling pathway. Knockdown of RCP in cancer cell lines inhibited TGF-α-induced MAPK activation and cell proliferation and migration [106].

HER2, an ErbB family member, is amplified in 15–20% of breast cancers and correlates with a poor prognosis. Antibody and small-molecule inhibitors of HER2 have been successfully used in the clinic. It is a unique member of the ErbB family in that it has no known ligand and instead acts as a heterodimerization partner for ligand-activated EGFR or HER3. The intracellular trafficking of HER2 is less well understood than for other RTKs, and there are contrasting reports as to whether HER2 is subject to endosomal recycling. One hypothesis suggests that HER2-containing dimers are maintained on the surface of the cell and are excluded from endocytosis, which accounts for the high surface distribution and its protection from EGF-induced downregulation. Another body of evidence supports a model in which HER2 undergoes rapid recycling, thus accounting for its almost exclusively cell surface localisation [107]. A recent study that supports the latter hypothesis was the observation that SORLA associates with HER2 and regulates its recycling back to the PM. Depletion of SORLA leads to the targeting of HER2 to lysosomes and sensitizes HER2-overexpressing cells to lysosome targeting drugs [47].

### 3.4. The Immune Checkpoint Protein PD-L1 Undergoes Endosomal Recycling

Programmed-death protein ligand 1 (PD-L1) belongs to the family of immune checkpoint proteins. It is a type I transmembrane protein that can be expressed on the surface of tumour cells. It possesses an extracellular N-terminal domain that interacts with the PD-1 receptor expressed on the surface of T cells. This interaction inhibits the activation, expansion, and effector functions of CD8^+^ T cells and is one mechanism that tumour cells have evolved to evade immune destruction [108]. The ratio between PD-1 and PD-L1 creates a balance between immune tolerance and autoimmunity. A disruption of this balance can lead to a number of disease states and has been reported to contribute to the onset of the auto-immune diseases arthritis and lupus [109].

PD-L1 has been reported to be expressed by 5–40% of tumour cells, which has led to massive efforts to develop therapies that disrupt its interaction with PD-1. Immune checkpoint inhibitors are monoclonal antibodies that work by binding to the extracellular domains of the checkpoint proteins and prevent them from binding their partner proteins. This prevents the ‘off’ signal from being sent to the T cells, allowing them to kill cancer cells. Immune checkpoint inhibitors have revolutionised oncology due to their long-term clinical benefit in a subset of patients and are arguably the most important development in cancer treatment in the last decade. There are currently seven approved immune checkpoint inhibitors, four of which target PD-L1 [110]. Along with this, there are hundreds of ongoing clinical trials throughout the globe that involve checkpoint inhibitors. The market for this class of drug reached $22 billion in 2019 and is forecast to reach $40 billion by 2025.

PD-L1 has recently been reported to undergo endosomal recycling under the control of a previously uncharacterised transmembrane protein called CMTM6. CMTM6 associates with PD-L1 at the cell surface and in REs, where it prevents PD-L1 from being ubiquitylated and targeted to lysosomes for degradation. Its depletion leads to reduced total PD-L1 levels by diverting PD-L1 from the recycling pathway to the degradative pathway, where it will ultimately be degraded in lysosomes. This leads to a reduction in the ability of tumour cells to downregulate T cell activity both in vitro and in vivo [111,112]. These findings suggest that selective inhibition of the endosomal recycling pathway may be an alternative means to downregulate PD-L1 and thus sensitize tumour cells to immune suppression.

## 4. Strategies for Inhibiting Endosomal Recycling

Given that the endosomal recycling pathway is centrally involved in a wide range of important human diseases, including cancer, neurodegenerative disorders, and pathogenic infections, it would seem obvious that developing therapeutics to target this pathway has potential to offer significant clinical benefit. For example, despite their widespread use, immune checkpoint inhibitors suffer from low response rates (~13% of all cancer patients, which increases up to approximately 58% as a combination therapy in melanoma), and most responders will eventually develop acquired resistance. In theory, a small molecule inhibitor of endosomal recycling could be employed to divert internalised PD-L1 into the degradative pathway, thus reducing the amount of PD-L1 at the cell surface and downregulating the ability of the tumour cells to inhibit CD8^+^ T cells. Such inhibitors may be useful in treating tumours that do not respond to antibodies targeting PD-L1 at the cell surface.

However, given that endosomal recycling is a universal physiological process, it might be expected that there would be significant toxicities associated with disrupting it. Inhibitors of the retrograde membrane trafficking pathway offer a precedent for the use of inhibitors of intracellular trafficking pathways, as they produce very few toxicities, even at high doses [67]. Furthermore, the endosomal recycling pathway appears to be hyperactivated in many cancers; thus, a therapeutic dose of inhibitor may have a negligible effect on non-malignant cells. Primaquine, an 8-aminoquinolone that can inhibit endosomal recycling, has been used for decades to treat malaria and is a safe and readily available drug. Discovery of inhibitors that block specific protein–protein interactions that are dysregulated in cancer, or that only occur in pathogen-infected cells, may also minimise toxicity.

The endosomal recycling pathway represents a novel and underexploited target in the area of biomedicine. To our knowledge, no therapeutics have been approved, or are currently in clinical trials, that have been specifically developed to target the recycling pathway. It is a widely held view that innovation in drug discovery is hampered by the pharmaceutical industry focusing their efforts on too few drug targets, leading to the development of ‘me too’ drugs with only marginal, if any, improvements in efficacy. We believe that the endosomal recycling pathway has huge potential as a target for developing novel first-in-class drugs that are likely to complement the therapies currently in use in the clinic.

Strategies that have been employed to inhibit the endosomal recycling pathway include inhibition of microtubule formation, prevention of endosomal acidification, or direct inhibition of key regulators of endosomal recycling with small molecules or inhibitory peptides.

### 4.1. Small-Molecule Inhibitors of Recycling

Small molecules are organic compounds that typically have a molecular weight of less than 900 Da. Due to their small size, they can easily diffuse across the plasma membrane, allowing them to interact with the cytoplasmic domain of cell-surface receptors or with intracellular signalling molecules. Plinabulin (BPI-2358) is a small molecule that disrupts endosomal recycling by inhibiting the polymerization of microtubules. It is currently being tested as a combination therapy in two clinical trials for advanced NSCLC. Recent work indicates that plinabulin exerts its anti-tumour effects by rapidly sequestering KRAS in endosomal vesicles, preventing it from activating the PI3K/Akt signalling cascade [113].

5-Benzylglycinyl-amiloride (UCD38B) is another small-molecule anti-cancer drug that has been reported to disrupt endosomal recycling. It belongs to a class of compounds that is being investigated as a potential treatment for high-grade glioma, a proliferative and deadly brain cancer characterized by the formation of avascular necrotic tumour domains and increased expression of proteins in the urokinase plasminogen activator system (uPAS). The main components of the uPAS system are the urokinase-type plasminogen activator (uPA), plasminogen activator inhibitor-1 (PAI-1), plasminogen activator inhibitor-2, tissue-type plasminogen activator, and the uPA receptor, all of which play a central role in inflammation, matrix remodelling during wound healing, tumour invasion, angiogenesis, and metastasis. UCD38B-mediated anti-tumour effects are due its ability to induce endocytosis and cause endosomes containing uPAS components to ‘mis-traffic’ to the perinuclear region of the cell, preventing their recycling back to the PM [114].

Primaquine is an aminoquinolone that has been used for decades as an anti-malaria drug. It is believed to interfere with the mitochondrial function of the *Plasmodium* parasite [115]. Primaquine is a weak base, and in human cell lines, it accumulates in endosomes where it neutralises the pH and strongly inhibits endosomal recycling [116]. It belongs to the same class of drugs as chloroquine and its derivative hydroxychlorquine, which have been widely tested in preclinical and clinical trials as an anti-cancer drug. There is some evidence to suggest that primaquine also has anti-tumoral activity [117] and sensitizes drug-resistant cancer cells to chemotherapies [118,119]. Our group has also found that primaquine has anti-cancer properties. It synergises with targeted therapies for breast and lung cancer cells and exerts these effects by inhibiting the endosomal recycling pathway (unpublished data).

### 4.2. Discovery of Novel Endosomal Recycling Inhibitors

Since the aberrant activity of Rabs has been implicated in a wide range of diseases, they are attractive targets for drug discovery. However, small GTPases are considered ‘undruggable’ given their lack of deep hydrophobic pockets that would allow binding of small molecules and their very affinity for GTP. Despite more than three decades of research, it is only very recently that a class of covalent inhibitors for one mutant form of KRAS has been discovered. Further complicating matters is the structural and functional similarity between the Rab family members, which makes it extremely challenging to identify Rab-specific inhibitors. Greater success may be achieved by indirectly inhibiting Rab function, rather than by searching for small molecules that directly target the Rab. Such strategies include modulating Rab-membrane association, targeting their gene expression, or inhibiting protein–protein interactions (PPIs) such as Rab–GEF or Rab–effector interactions [120]. Since Rab effectors have more structural diversity, they are good candidates for direct targeting with small molecule inhibitors.

At least two selective inhibitors of Rab GGTase have been identified, one of which inhibits the prenylation of Rabs in human myeloma cells, induces apoptosis, and improves survival in mouse models of multiple myeloma [121]. Blocking the binding of GTP, by disrupting the interaction between a Rab and its GEF, would reduce or abolish the activity of the targeted Rab. Similarly, inhibiting the interaction of a Rab with its downstream effector protein would also modulate its activity. Stapled peptides have been used successfully to disrupt the binding of Rab8 with its effector OCRL1 [122] and between Rab25 and RCP [123]. The Rab25 targeting peptide, RFP14, preferentially binds to Rab25 over Rab11a and inhibits cell proliferation and migration in the breast cancer cell lines in which Rab25 is oncogenic, but augments these phenotypes in the cell lines in which it is tumour-suppressive [123].

The expression of Rabs and their regulatory proteins are frequently modulated by microRNAs (miRNA). For example, miR-338-3p inhibits Rab14 gene expression and has tumour suppressor properties [124]. The delivery of synthetic miRNAs with tumour-suppressive effects may block cancer development. Conversely, antagomiRs (small molecule miRNA inhibitors) could inactivate miRNAs with oncogenic properties. The biotech industry has invested heavily in miRNA-based drugs, and a number are currently in clinical trials.

### 4.3. Exploiting the Endosomal Recycling Pathway for Drug Delivery

The endosomal recycling pathway has also been exploited as a means for delivering anti-cancer drugs inside cells, especially as a means for crossing the blood–brain barrier (BBB). Iron is important for brain function, and iron-loaded transferrin binds to the transferrin receptor on the surface of microvascular endothelial cells. This complex is transcytosed from the blood to the brain side of the barrier. It is believed that this is a promising route for delivering drugs into the brain of patients with neurological disorders; however, despite decades of research to develop transferrin receptor (TfR) binding antibody-drug conjugates, there has been little clinical success [125]. TfR is overexpressed in a number of cancers since they require more iron than normal cells to sustain their rapid proliferation. A diphtheria toxin-transferrin conjugate reached phase III clinical trials investigating it as a treatment for glioblastoma multiforme [126]. A phase I/II clinical trial is currently recruiting to test a modified TfR-binding antibody-drug conjugate in patients with metastatic or locally advanced unresectable solid tumours (NCT03543813; ClinicalTrials.gov).

## 5. Summary

Despite the importance of the endosomal recycling pathway for normal physiological and pathological processes, it remains relatively understudied in comparison to other membrane trafficking pathways. Further study is required to unravel the signals that determine whether a cargo is to be degraded or recycled. It will be necessary to decipher the context that determines whether an alteration to a regulator of endosomal recycling is oncogenic or tumour-suppressive. We need improved assays for studying endosomal recycling in vitro and ways for examining it in vivo. Other questions that remain include: How many recycling pathways are there in a cell? Do cargoes have dedicated recycling machineries? How is each regulated and what are the mechanisms of crosstalk between them? How do the various membrane trafficking pathways communicate with each other, and what level of overlap is there between them? Can modulating endosomal recycling be beneficial in the clinic? Answering these questions will provide greater understanding of the normal physiological role of the endosomal recycling pathway, will lead to insights into the pathology of diseases and host–pathogens interactions, and identify new targets for developing novel therapeutics.

## Figures and Tables

**Figure 1 ijms-21-06074-f001:**
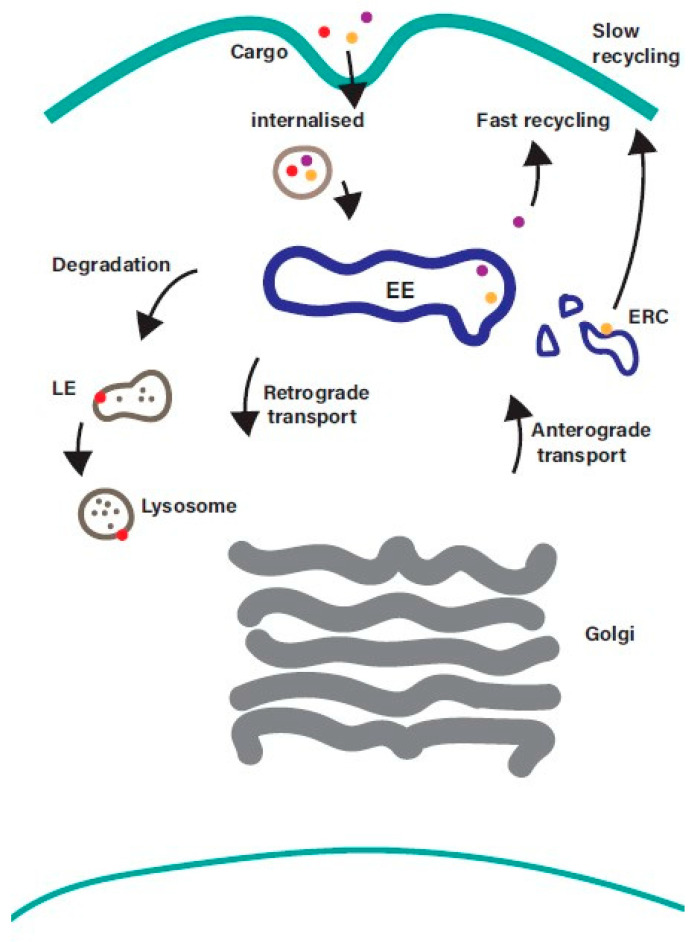
Overview of the endosomal recycling pathway. Cargo internalised from the cell surface by clathrin-mediated endocytosis or clathrin-independent endocytosis converges at the early endosome, where it is sorted for onward transport along the recycling, degradative, or retrograde trafficking pathways. Cargo returning to the plasma membrane can be recycled directly from the early endosome (fast recycling pathway) or indirectly from the endosomal recycling compartment (slow recycling pathway). EE—early endosomes; LE—late endosomes; ERC—endosomal recycling compartment.

**Figure 2 ijms-21-06074-f002:**
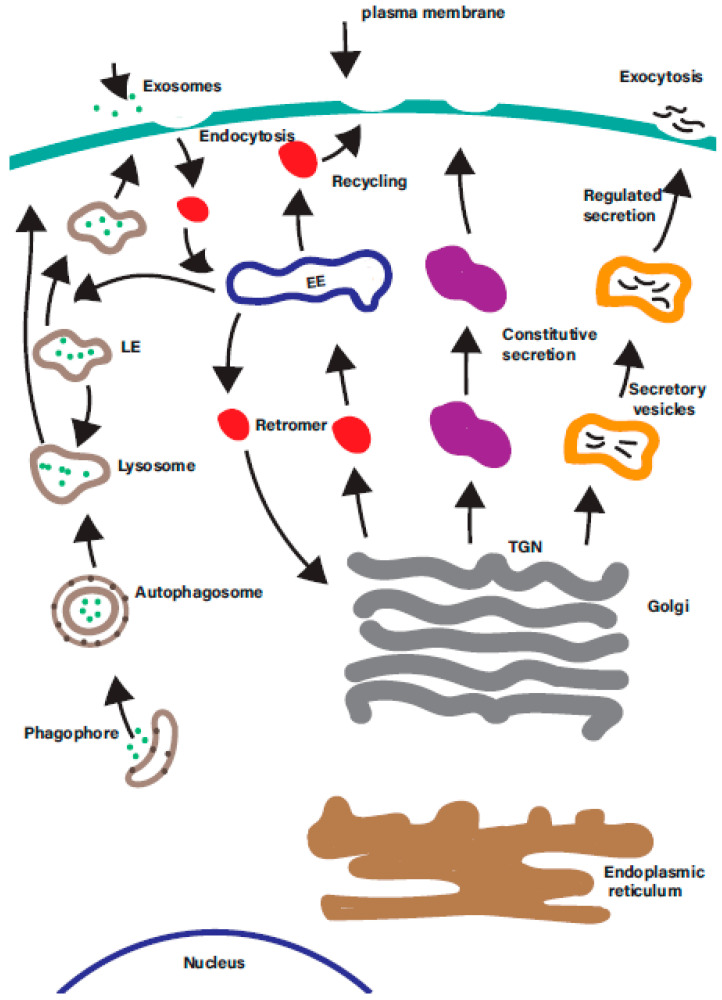
There are a multitude of membrane trafficking pathways in the cell. A simplified overview of the many overlapping intracellular membrane trafficking in the cells. Each is regulated by one or more organelle-specific Rab GTPases. Membrane trafficking pathways indicated include the endocytic, degradative, secretory, retrograde, endosomal recycling, and autophagic pathways. EE—early endosomes; LE—late endosomes; ERC—endosomal recycling compartment; TGN—*trans*-Golgi network.

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
