# Peer review of "The Endosomal Recycling Pathway—At the Crossroads of the Cell"

_ijms, 2020, doi:10.3390/ijms21176074_

Round 1

Reviewer 1 Report

Dr. Mary J. O’Sullivan et al. Reported a review article of endosomal recycling pathway. The article is well written and available for publication. Some minor comments were advised.

  1. In reference section, a double reference number is found.
  2. Some small grammar error is advised.

 Abstract section (line 19)

This review will provide an overview of the normal physiological role of the endosomal recycling pathway, it will describe…...

è This review will provide an overview of the normal physiological role of the endosomal recycling pathway, “and” it will describe…...

Some error typing words

[1] page 8, line 296

integrin and drives invasive migration by directing “??” integrin and EGFR..

[2] page 11, line 410

It has a number of ligands including EGF, TGF-“??”, HB-EGF and epiregulin

[3] page 11, line 425-430

TGF-“??”

Author Response

We thank Reviewer 1 for their comments.

  1. We could not find a duplicated reference.
  2. “and” has been included in the abstract.
  3. Page 8, Line 296. The font of the relevant characters has been changed to Symbol.
  4. Page 11, Line 410. The font of the relevant characters has been changed to Symbol.
  5. Page 11, line 425-430. The font of the relevant characters has been changed to Symbol

Reviewer 2 Report

The presented manuscript is a very interesting review. I believe it will be appreciated by readers. The text is logically arranged, easy to read and understand. I have no recommendations to edit the text. I recommend to accept the manuscript for publication.

Author Response

We thank Reviewer 2 for their comments and note that they are happy with the manuscript as submitted. 

Reviewer 3 Report

In this review O`Sullivan and Lindsay discuss the role of endosomal recycling pathway in cell physiology and pathological developments. The authors give a brief overview of endocytic pathway and then mainly focus on edosomal recycling.  The authors discuss in detail various pathological disorders developed mainly due to dysregulation of endosomal recycling and/or endocytosis. In the end the authors discuss the potential therapeutic targets for the pharmacological intervention of the pathology and the available pharmacological strategies/small molecule inhibitors for clinical use.

Suggestions:

2.3 MVID: A pathologically less sever variant of MVID is caused mutations in Stx3 gene which is also involved in endocytosis. That may also be discussed in this section.

Figure legends may be improved to be more elaborative. Please include abbreviations in the legends.

Author Response

We thank Reviewer 3 for their comments and have included all their suggestions in the revised manuscript. 

Lines 252 - 256: We have included two sentences in the MVID section that discuss the discovery of mutations in Stx3 in two patients that presented with MVID symptoms, and describe how myosin Vb and syntaxin 3 are functionally linked.

Lines 59 - 64; 91 - 96: We have expanded the figure legends and included the abbreviations.